# Observation of magneto-electric rectification at non-relativistic intensities

M. Tuan Trinh[1,2 ✉], Gregory Smail[3], Krishnandu Makhal[1], Da Seul Yang[4], Jinsang Kim [4] & Stephen C. Rand[1,3]

The subject of electromagnetism has often been called electrodynamics to emphasize the dominance of the electric field in dynamic light–matter interactions that take place under non-relativistic conditions. Here we show experimentally that the often neglected optical magnetic field can nevertheless play an important role in a class of optical nonlinearities driven by both the electric and magnetic components of light at modest (non-relativistic) intensities. We specifically report the observation of magneto-electric rectification, a previously unexplored nonlinearity at the molecular level which has important potential for energy conversion, ultrafast switching, nano-photonics, and nonlinear optics. Our experiments were carried out in nanocrystalline pentacene thin films possessing spatial inversion symmetry that prohibited second-order, all-electric nonlinearities but allowed magneto-electric rectification.

[1] Center for Dynamic Magneto-Optics, Dept. of Electrical Engineering & Computer Science, University of Michigan, Ann Arbor, MI 48109, USA. [2] Department of Physics, University of South Florida, Tampa, FL 33620, USA. [3] Division of Applied Physics, University of Michigan, Ann Arbor, MI 48109, USA. [4] Department. of Materials Science, University of Michigan, Ann Arbor, MI 48109, USA. ✉email: tm4@usf.edu

Throughout most of the history of electromagnetism and nonlinear optics, reports of scientific discoveries have been dominated by strong electric-field interactions and relatively weak dynamic magnetic effects[1]. For a century after the time of James Clerk Maxwell, available light sources were too weak to elicit more than a linear parametric response from optical materials and in agreement with theoretical expectations[2] magnetic response at optical frequencies was found to be negligible. Since the advent of the laser, however, and the invention of chirped pulse amplification in particular[3], available intensities have reached relativistic levels ($I \sim 10^{18}$ W/cm$^2$) at which, as is well-known[1], magnetic effects compare well with electric response. In the relativistic regime, charges are accelerated to light speed on timescales less than the period of light and the magnetic Lorentz force rises to match the force of the electric field. At these ultrahigh intensities, nonlinear dynamics finally allow the optical magnetic field to assert itself in the production of large magnetic fields[4], charge drift[5], ponderomotive forces[6], and the like. Unfortunately, no optical interaction to date has been capable of exploiting the Lorentz force of light to control magnetic dispersion or charge separation[5] under non-relativistic conditions, or switch light at right angles, or demonstrate other magnetic effects that become possible with dual electromagnetic field interactions[6]. Here, we report optical rectification mediated by the Lorentz force in a centrosymmetric medium under non-relativistic conditions.

As early as 1926, Debye speculated that phenomena jointly dependent on electric ($E$) and magnetic ($H$) fields of electromagnetic waves in the form $EH$ could take place in nature[7], but no magneto-electric (M-E) effects at the atomic level were reported for the next eighty years[8]. As a result, alternative approaches have been devised to elicit magnetic response through metamaterial design and "bulk" M-E effects have been explored in multiferroic materials to realize functionality that is both magnetic and dynamic[9,10]. Very recently, however, it was reported that magnetic properties of homogeneous dielectric media can be controlled by nonlinearities driven jointly by electric and magnetic field components of light at the atomic level[11–13], as envisioned by Debye. Such nonlinear M-E interactions have been shown to drive bound electrons on curved trajectories very effectively under non-relativistic conditions via an optical torque mechanism[12,13]. The resulting curved motion of bound charges in space breaks temporal and spatial inversion symmetry, as dictated by parity-time (P-T) symmetry[14] rather than the medium symmetry, giving rise to new physical phenomena. P-T symmetry has led in recent years to discoveries of magnetic topological insulators[15], non-reciprocal optics[16], the quantum spin Hall effect of light[17], and topologically-protected edge states in photonic technology[18,19]. Consequently, it should not be surprising that this unusual symmetry holds unforeseen consequences for dual field interactions too, such as longitudinally-polarized second harmonic radiation, induced radiant magnetization at optical frequencies, and the longitudinal optical charge separation (rectification) reported here.

While experimental results have been published on magnetization induced by light at the molecular level[8,13], so far there are no experimental reports of M-E rectification. Although M-E rectification is a quadratic nonlinearity, no special crystal symmetry is necessary for it to take place. This is contrary to conventional nonlinear optics, which requires a lack of inversion symmetry for second-order processes, but is in accord with the combined lack of parity or time symmetry exhibited by the M-E interaction Hamiltonian[18]. Hence the role of P-T symmetry is confirmed in the present work, since we report the observation of dynamic M-E rectification in a centrosymmetric solid. Here, we point out the advantages of this interaction for enabling ultrafast all-optical switching with unique geometry

and in providing novel sensing technology, as well as applications such as M-E energy conversion and terahertz emission in unbiased materials that are simply not possible with all-electric nonlinear interactions. Our results also demonstrate that relativistic optical response is attainable under non-relativistic conditions.

## Results

Our experiment utilized a crossed beam, pump-probe approach to investigate nonlinear M-E rectification (MER) in a thin film of the organic semiconductor pentacene, prepared by thermal deposition on a glass substrate in vacuum. We chose pentacene because the random nanocrystalline structure of the thin film rendered it effectively centrosymmetric, thereby prohibiting all-electric rectification. The proximity of the lowest energy resonance of pentacene to our excitation laser wavelength also permitted significant enhancement of the ME coupling. Fig. 1c presents the spectrum of the pentacene thin film showing its absorption peak at 670 nm. The sample, sealed in an inert gas environment to avoid photo-oxidation during measurement or storage, was positioned at the intersection of the pump and probe beams. The thin film had a thickness of 400 nm, which minimized pulse dispersion in the sample. It consisted of close-packed crystallites of diameter ~100–200 nm with random orientations, making it incapable of supporting all-electric second harmonic generation (SHG) in the bulk, see supporting information (SI) for the sample characterization. On the other hand a harmonic signal was expected to arise irrespective of pump-probe delay due to broken symmetry at the two surfaces of the thin film whenever the probe field had components perpendicular to the surface. In addition, harmonic radiation was expected to be induced by broken inversion symmetry of the medium when the pump and probe waves were coincident. Presence of a quasi-static MER field removes isotropy of the sample in the interaction volume, thereby enabling second harmonic generation[20,21]. Thus second harmonic generation from the probe should be induced only during the M-E interaction caused by the pump pulse, producing a transient signal that lasts only as long as the induced rectification field. We refer to the corresponding signal as ME-field-induced second harmonic or ME-FISH to distinguish it from electric-field-induced SHG, which has been used successfully to detect charge separation fields across interfaces[22] and in organic semiconductors[23].

Fig. 1a, b present the crossed-beam, pump-probe experimental setup for investigating ME-FISH using an amplified femtosecond laser system. In the presence of the pump pulse, the induced dipole field caused by magneto-electric rectification is parallel to the propagation axis of the pump beam. So in order to maximize the ME-FISH signal, the probe polarization needs to be aligned parallel to this direction (Fig. 1b). With beams crossing at 90° this requirement is easily satisfied, but the crossed-beam geometry has a drawback that pump and probe pulse-fronts ordinarily have a protracted period of spatial overlap in the sample, resulting in very poor temporal resolution of the interaction. To circumvent this, we tilted the pulse-fronts of both beams to 45°, picking the unique case shown in the inset of Fig. 1a, which maintains a temporal resolution limited by the pulse widths themselves when the phase planes of the pulses are perfectly flat. The vector field interaction can then be manipulated by controlling the pump and probe polarization via half-wave plates. This configuration permitted SHG signals of magneto-electric origin (ME-FISH) to be clearly distinguished from surface second harmonic (SSHG) of all-electric origin by rotating the probe polarization to various angles $\alpha$ with respect to the $y$ axis in the $y$–$z$ plane. Details of the experimental approach needed to minimize the instrumental time

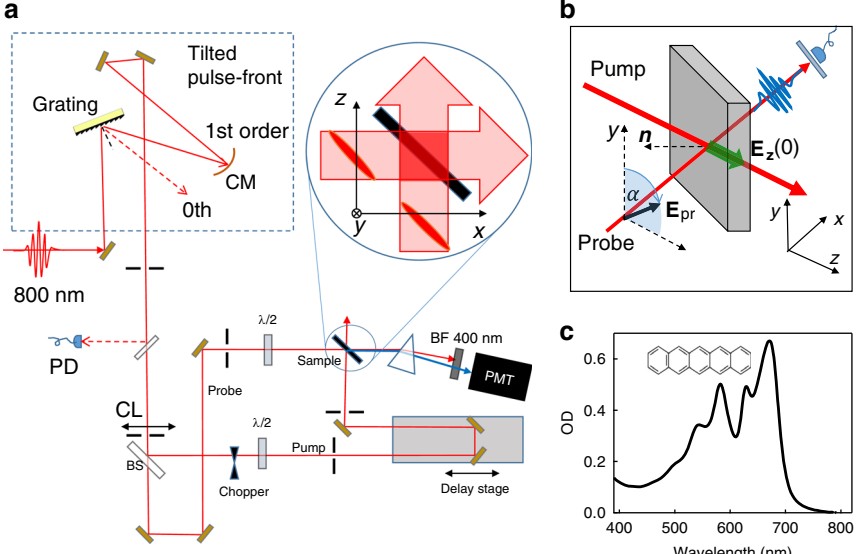

**Fig. 1 Experimental setup and sample characterization. a** Schematic of the crossed beam pump-probe experiment for detection of magneto-electric rectification. The dashed rectangle depicts a controller for tilting the pulse-front that consists of a grating, a cylindrical concave mirror (CM), and a pair of flat mirrors. The first order laser beam from the grating passed through a cylindrical lens (CL). The CM and CL mirrors provided chirp and dispersion correction, respectively. The beam splitter (60/40) then split the incident laser beam into pump and probe pulses. Half-wave plates (λ/2) were used to control polarization. SHG signal was detected by a photomultiplier tube (PMT) through a prism and a bandpass filter (BF) centered at $\lambda = 400$ nm. A magnified top view of the sample region (circular inset) shows the tilted pump and probe pulses approaching the interaction region of the sample. A photodiode (PD) was used to synchronize the laser with the chopper. **b** A 3D view of the beam geometry. Probe polarization angle α is measured with respect to the y axis. **n** is the surface normal. The wide green arrow represents the forward pump-induced rectification electric field, $E_z(0)$. **c** Absorption spectrum of the pentacene sample, showing the absence of absorption at 800 nm and weak absorption at the 2-photon absorption wavelength of 400 nm. OD = optical density.

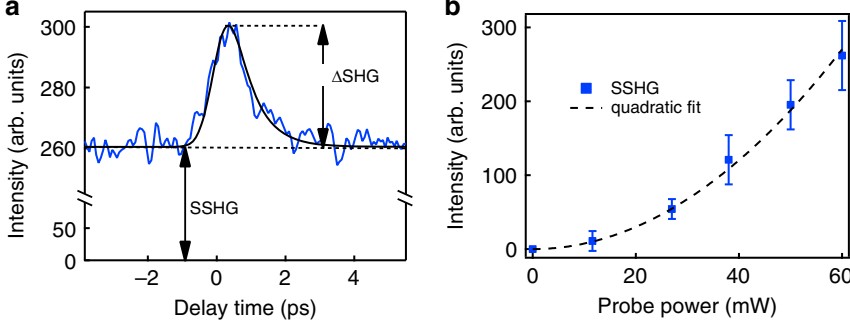

**Fig. 2 Second harmonic probe signal from a pentacene thin film. a** SHG intensity as a function of pump-probe delay, with the probe polarization angle set at 60° with respect to vertical. Two distinct contributions to the signal may be noted—a constant background (labeled SSHG), and a pump-induced signal at zero delay (labelled ΔSHG). The solid black curve is the convolution of instrumental response and simulation of the MER response based on the torque model[12]. **b** The background SSHG signal varies quadratically with input probe power as expected (dashed-curve). The error bars were determined by the variance of the signal before time zero.

resolution to enable these measurements are given in the section on Methods and Supporting Information.

We conducted two sets of experiments using the apparatus of Fig. 1a. First, the relative timing of the pump and probe pulses was varied with a mechanical delay stage to search for overlap of the pump and probe pulses. The position of zero relative delay was determined by placing a GaAs wafer in the sample plane and adjusting the stage to observe the maximum sum frequency signal in reflection (see Supporting Information). With the sample in place, the delay stage was then swept over a range of 10 ps. The results are shown in Fig. 2a for SHG signal intensity vs. time and in Fig. 2b for SHG signal intensity versus incident probe intensity. In Fig. 2a, a strong background attributable to SSHG may be noted at all delay times. The quadratic power dependence of the probe background signal on probe power in Fig. 2b confirms that

the observed signal is second harmonic radiation. However, because this signal is constant versus delay it is clearly not pump-induced. In Fig. 2a on the other hand, a peak appears at zero delay time, which is evidently pump-induced.

Both classical and quantum theoretical models of MER have shown that at suitably high (non-relativistic) intensities, optical electric, and magnetic fields drive bound charges away from their nuclei by a mechanism that involves both the Lorentz force and magnetic torque[12]. This is the process that results in an internal magneto-electric rectification field $E_z(0)$ proportional to the nonlinear polarization $P_z^{(2)}(0)$ described by[24]

$$P_z^{(2)}(0) = (1/c)\chi_{zyx}^{(\text{eme})}(0; -\omega, \omega)H_y^*(-\omega)E_x(\omega) = (\epsilon(0) - \epsilon_0)E_z(0)$$

$$(1)$$

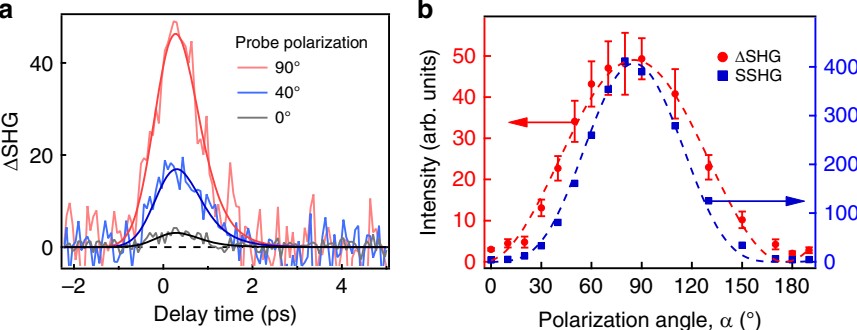

**Fig. 3 Probe polarization dependence of the second harmonic signal. a** Pump-induced ME-FISH ($\Delta$SHG) signals from the pentacene thin film for three different probe polarization angles. At 0º polarization, the $E$ field is vertical, parallel to the sample surface. At 90º polarization, the $E$ field is horizontal, parallel to the propagation axis of the pump. **b** Polarization angle dependence for surface SHG (squares) and pump-induced SHG (circles). The dashed curves are the fits with $\sin^4(\alpha)$ and $\sin^2(\alpha)$ functions for SSHG and $\Delta$SHG, respectively. The error bars were determined by the variance of the signal before time zero.

where $\mathbf{E}(\omega)$ and $\mathbf{H}(\omega)$ are the electric and magnetic fields of the pump pulse and $E_z(0)$ is the zero frequency field produced by the MER process in the $z$ direction. $\epsilon_0$ and $\epsilon(0)$ are the permittivities of vacuum and of the medium at zero frequency, respectively. $c$ and $\chi$ are the speed of light and a susceptibility matrix element, respectively. In the presence of the static internal field $E_z(0)$, a DC Kerr effect can take place, which is all-electric in nature, resulting in the four-wave mixing polarization

$$P_z^{(3)}(2\omega) = \epsilon_0\chi_{zzzz}^{(3)}(2\omega; 0, \omega, \omega)E_z(0)E_0^2(\omega) \quad (2)$$

where $E_0$ is the incident probe field amplitude. The third-order polarization in Eq. (2) is the origin of frequency-doubled radiation that propagates parallel to the probe, producing an induced harmonic signal, which is detected in our experiment at $\lambda = 400$ nm.

Fig. 3 shows the pump-induced second harmonic signal intensity as a function of pump-probe delay for several values of the angle $\alpha$ between the $y$ axis and the probe electric field. In Fig. 3a the pump-induced peak at zero delay is clearly largest at $\alpha = 90°$, when the probe electric field aligns with the pump-induced electric field $E_z(0)$, (see also Fig. 1b). On the other hand the induced signal is negligible when probe polarization is orthogonal to $E_z(0)$ at $\alpha = 0°$. This variation can readily be compared with expectations for a magneto-electric interaction (ME-FISH) or conventional SSHG. Fig. 3b presents such a comparison taking into account the 90° crossed-beam geometry and the orientation of the sample plane at 45° with respect to both input beams. In the case of SSHG, only the probe field component, which is perpendicular to the sample surface generates second harmonic signal. This component is $(E_\perp)_{probe} = E_0\cos(45°)\sin(\alpha)$, yielding the harmonic signal $S(2\omega) \propto (P_\perp^{SSHG})^2 \propto I_0^2\sin^4(\alpha)$. In the case of ME-FISH, polarization components perpendicular to $E_z(0)$ preserve centrosymmetry. Consequently, in this case inversion symmetry is broken (and harmonic generation thereby allowed) only for the polarization component $P_z^{(3)}$ oriented along the $z$ axis. From Eq. (2) it can be shown that this component is angle-independent: $P_z^{(3)} \propto E_0^2(\omega)$, see SI. Hence, the second harmonic polarization is the sum of surface and magneto-electric terms, with a total signal intensity of

$$S_{tot} \propto \left|P_\perp^{SSHG} + P_z^{(3)}\right|^2 = \left|P_\perp^{SSHG}\right|^2 + 2\left|P_\perp^{SSHG} \cdot P_z^{(3)}\right| + \left|P_z^{(3)}\right|^2 \quad (3)$$

The first term on the right side of Eq. (3) corresponds to the background, which is independent of pump-probe delay. As discussed above, its dependence on probe polarization should be $S(2\omega) \propto \sin^4(\alpha)$. The second term is induced by the rectification

field and varies in direct proportion to $P_\perp^{SSHG} \propto \sin^2(\alpha)$, as $P_z^{(3)}$ is independent of $\alpha$. The pump-induced (ME-FISH) signal should therefore vary as $S \propto \sin^2(\alpha)$. The third term is of higher order with respect to the fields. Hence, it is small and has no angle-dependence, so it plays no further role in our analysis. We note that excellent agreement is obtained in Fig. 3b for a $\sin^4(\alpha)$ fit to the background signal, whereas a $\sin^2(\alpha)$ dependence is required to fit the induced signal variation with polarization angle. Consequently, this provides strong evidence that the ever-present background in our data results from SSHG, whereas the induced harmonic signal is indeed magneto-electric in origin (ME-FISH).

In Fig. 3a, a fit to the ME-FISH signal that convolves instrumental response and the torque model[12] of M-E rectification results in a risetime of ~0.5 ps and a decay time constant of $\tau$ ~0.6 ps. The response curve is somewhat asymmetric and both the rise and fall times exceed the temporal resolution of the setup. The prolonged risetime is only reproduced when magneto-electric torque dynamics are included in our simulations, providing some indication that torque may add to parametric resonance to produce enhanced Lorentz force response, just as in simple molecular liquids (See the simulations in Supporting Information). The rise and fall times of ME signal depend on the material properties such as damping constant and rotation/libration frequency as well as on experimental conditions such as the pulse duration. The decay time of the ME-FISH signal of ~0.6 ps is slow compared to characteristic times for pentacene dynamics, such as the 80 fs for singlet exciton fission process[25,26]. However, it is much faster than the fluorescence lifetime of pentacene, which is reported to be ~24 ns[27] and may reflect the timescale of electron librational relaxation taking place in the sample during the dynamics initiated by the magneto-electric process[13].

An internal field such as $E_z(0)$ is required to break the bulk inversion symmetry in order for frequency-doubling to be possible in nanocrystalline pentacene. We note that pentacene is not ferroelectric and at the wavelength of our experiment no charges are liberated by electronic excitation by the laser (see SI). The pump and probe energies in our experiments are both 1.55 eV, well below the LUMO energy of pentacene (see Fig. 1c), so electron-hole pair production is negligible and does not contribute to internal fields (see SI). Furthermore, Fig. 4a, b, respectively, indicate that the induced signal itself is a quadratic nonlinearity that displays no significant dependence on pump polarization. Consequently the induced harmonic signal is not the result of hyper-Rayleigh scattering or surface second harmonic generation of the pump itself. Both of these effects would vary strongly with pump polarization in

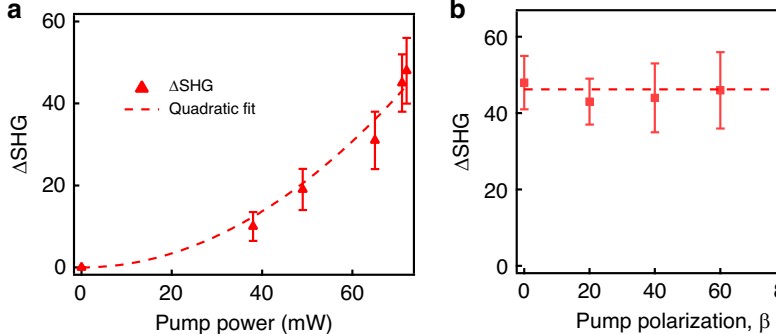

**Fig. 4 Pump power and pump polarization dependence. a** Quadratic pump power dependence of pump-induced SHG. The dashed curve is a quadratic fit. The relatively large error bars reflect the fluctuations of the pump-induced SHG signal, relatively weak in comparison with the surface SHG. **b** The pump-induced SHG as a function of pump polarization, β. The dashed-line is a horizontal line fit.

the lab frame. The induced second harmonic also cannot arise from an electric quadrupole interaction. Rectification is not produced by quadrupole interactions (see SI) and any harmonic radiation of quadrupolar origin would contribute to the background signal but not to a pump-induced change of the probe harmonic signal (ΔSHG). Our results are therefore uniquely consistent with a magneto-electric mechanism since only the combined effect of electric and magnetic optical fields can explain a second-order longitudinal rectification field in samples possessing inversion symmetry. Furthermore, only a MER field has the right orientation to explain harmonic generation that is independent of pump polarization, as it points along the propagation axis of the pump wave[11,12].

Unlike previous experiments on electric-field-induced harmonic generation[23], the internal field that induces SHG in the present work is of optical origin and results from a second-order nonlinearity that takes place in polycrystalline pentacene. The fact that our samples are effectively centrosymmetric and yet respond with a quadratic dependence on pump power rules out an all-electric origin for the induced second harmonic generation. As argued above, the excellence of the fit to probe polarization dependence in Fig. 3b based on the magneto-electric polarization in Eq. (3), and the absence of pump polarization dependence in Fig. 4b, furnish compelling evidence that the ME-FISH mechanism is responsible for the results in Figs. 2a and 3a.

Magneto-electric rectification involving the optical magnetic field would ordinarily require relativistic light intensities to make up for the weakness of the Lorentz force. Hence, our results indicate that an enhancement mechanism must exist to lower the intensity requirement. Such a mechanism has been analyzed previously, and independent experimental evidence for it has been published based on light scattering experiments[13,28,29]. The present findings extend these results by furnishing not only the report of magneto-electric rectification but also indirect confirmation that an enhancement mechanism applies to static ME field generation as well as optical magnetization with non-relativistic fields. Lacking a theory specifically applicable to solid materials we surmise that enhancement occurs universally by a combination of parametric resonance and magnetic torque dynamics as outlined in earlier work on simple molecular liquids[12,29]. This has interesting implications for direct conversion of light energy into static voltages at moderate optical intensities ($\sim10^9$ W/cm$^2$)[11], as our experiments were performed with intensities, which are nine orders of magnitude below the relativistic threshold ($\sim10^{18}$ W/cm$^2$). We note that theoretically the magneto-electric rectification field, which mediates the ME-FISH process may be driven as

effectively with incoherent light as coherent light[11], meaning that M-E interactions could support the direct conversion of sunlight to electrical energy in insulators as previously conjectured. Light-by-light switching at an angle of 90° should also be possible in novel magneto-photonic devices.

## Methods

**Experimental setup**. The experimental setup (Fig. 1a) consists of a custom, amplified femtosecond laser, which delivers pulses of 0.4 mJ at a 10 kHz repetition rate (Amplitude Laser, Inc.). Pulse durations were electronically tunable from 20 to 500 fs, but for this experiment were left fixed at 100 fs with a center wavelength of 800 nm. The incident laser was split into two beams. One of these beams passed through a delay stage to control relative timing between pump and probe pulses. Its intensity was adjusted with a rotatable half-wave plate followed by a polarizer. The polarizations of the pump and probe beams were also adjusted using rotatable half-wave plates. A bandpass filter (10 nm bandwidth at 400 nm) was placed after the sample in the path of the probe to transmit only the second harmonic signal for detection by a photomultiplier. To subtract the static background in order to study pure pump-induced signal, a mechanical chopper was used to modulate the pump light at 1 kHz. The pump-induced signal was measured as signal (ON)–signal (OFF) controlled by a LabVIEW program. To synchronize the laser frequency with the chopper, a fast photodiode was used for triggering. The pump diameter at the sample position was 1.5 mm, however, the laser spot on the sample was an elliptical shape with axes of 1.5 and 2.1 mm because the incident angle was 45°. For 100 fs pulses, our maximum pump pulse intensity on the sample was therefore $\sim7 \times 10^8$ W/cm$^2$. To detect a pump-induced longitudinal rectification field, a crossed-beam geometry was used. This configuration permitted the observation of frequency-doubling of probe light enabled by the loss of inversion symmetry accompanying the appearance of the rectification field. It also permitted SHG signals of magneto-electric origin to be distinguished from those of all-electric origin through rotation of the probe polarization. When the probe polarization was rotated to lie parallel to the axis of propagation of the pump, harmonic generation was maximized as shown in Fig. 3a. In this orientation the probe field was sensitive to broken symmetry much like extraordinary polarization in a uniaxial crystal. When it was perpendicular to the axis, the SHG signal vanished because inversion symmetry was not broken perpendicular to the induced field axis, much like ordinary polarization in a uniaxial crystal.

**Sample preparation and characterization**. The pentacene films were deposited on glass by vacuum thermal evaporation at a deposition rate of 0.1 nm/s under the base pressure of $3 \times 10^{-7}$ Torr. The thickness of the sample in this study was 400 nm and individual crystallites in the film had diameters of ~100–200 nm. During storage and measurements, the samples were kept in an inert gas environment to avoid photo-oxidation. The sample surface was characterized by atomic force microscopy (AFM; Asylum Research MFP-3D AFM). AFM images were recorded under tapping mode with a CT300-25 Aspire probe (See Supporting Information, section 6). The absorption spectrum of the pentacene thin film was measured using an UV-3600 Plus spectrophotometer (Shimadzu).

## Data availability
The data are available from the corresponding authors upon reasonable request.

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

## Acknowledgements
This research was supported by the MURI Center for Dynamic Magneto-optics, the Air Force Office of Scientific Research (FA9550-12-1-0119 and FA9550-14-1-0040); DURIP grant (FA9550-15-1-0307), and NSF grant (NSF1947070).

## Author contributions
S.C.R. an M.T.T. conceived the project and wrote the manuscript. M.T.T, G.S., and K.M. designed and constructed the experimental setup. M.T.T. conducted the experiments and analyzed the data. G.S. performed the simulations. D.S.Y. and J.K. provided and characterized the samples. All authors discussed and contributed to the scientific results and the manuscript.

## Competing interests
The authors declare no competing interests.
