## [Peer Review File · Nature Communications]

Reviewers' Comments:

Reviewer #1:

Remarks to the Author:

I attached my review as a pdf below.

Mark G. Kuzyk

Reviewer #2:

Remarks to the Author:

The authors propose a nonlinear mechanism arising from both the electric and magnetic field of laser light (EH). This second order nonlinear process is argued to result in magneto-electric rectification even though the magnetic field of laser light is weak for the intensities used. To explain the amplification, the authors make an analogy to molecular systems, where nonlinear magneto-electric interactions have been shown to drive bound electrons on curved trajectories via an optical torque mechanism.

The importance of this paper is that it proposes a possible way to transiently break temporal and spatial inversion symmetry by using an electromagnetic field, rather than breaking inversion symmetry, and thus generate a second order nonlinearity. In particular, a symmetry breaking static field is generated by the second order polarization via a second order frequency difference process that involves both the electric and magnetic fields of light. While such dynamical symmetry breaking process is expected to be weak, since the magnetic field strength is suppressed for the intensities used, the authors make an analogy to previous work in molecular systems to argue that such second-order process is enhanced.

The importance of the proposed effect is that it allows for control of dynamical symmetry breaking by using the laser electromagnetic field. In this way, new physical properties that are directly controlled by light may be expected, for example, in quantum materials, provided that the mechanism of amplification applies to such materials. The possibility for dynamical symmetry breaking via rectification by using the electromagnetic field propagating in thin films has received attention recently in superconductors and topological materials (see, for example, the works in Nature Photonics 13, 707–713 (2019), Phys. Rev. Lett. 124, 207003 (2020), etc). Another possibility for transient magnetization effects in condensed matter systems, photogenerated through coherent nonlinear processes during the laser pulse despite the laser's weak magnetic field, was suggested in Nature Physics 5, 517 (2009).

In the present experiment, SHG may be generated at the surfaces of the thin film, but the authors were able to identify two different signals with different behaviors, and thus distinguish between SHG and the proposed EH nonlinear effect that occurs during the pump pulse, as it is pump-induced. In particular, SHG from the probe field was observed to arise due to symmetry breaking by magneto-electric interaction induced by the pump pulse, which gives rise to a rectification field. In my opinion, the authors provide evidence that they observe a pump-induced second-order signal with characteristic polarization and time-delay dependence, which they attribute to the proposed dynamical symmetry breaking mechanism due to the pump electric and magnetic fields that generate a static electric field. The authors have presented evidence for their proposed mechanism, except for the reason for its enhancement despite the weak laser magnetic field, for which they make the analogy to their previous works in molecular systems. The authors should comment on this, i.e. is this amplification a general effect or does it apply to specific materials. If the latter is the case, which would make the participation of the laser magnetic field plausible, what is the important condition for obtaining the proposed amplification? Does a similar laser-induced dynamical symmetry breaking mechanism apply to quantum materials, which would allow one to coherently control their properties, or is it specific to nonlinear materials like pentacene? If

yes, what are the main conditions for amplification and what kind of nonlinearity is required? Could a similar dynamical symmetry breaking effect and light-induced symmetry breaking static field (rectification) occur in a highly nonlinear thin film material with inversion symmetry due to the electric field alone, as proposed e.g. in Nature Photonics 13, 707 (2019), Phys. Rev. Lett. 124, 207003 (2020)? What is the main evidence in the experiment that requires invoking the weak magnetic field of the laser despite the weak Lorentz force?

In conclusion, this paper proposes an all-optical symmetry breaking mechanism that may also open new ways to control quantum materials, if the authors can argue that it is widely applicable, i.e. amplification is possible under general conditions in condensed matter, as well as in molecular systems. Is there an estimate for what kind of nonlinearity of the centrosymmetric material is required for significant symmetry breaking? Is rotational motion as in molecular systems required to amplify the proposed effect and make it observable? What kind of magnetic interaction is required for the proposed effect in different systems? Is this specific to certain thin film systems, and if yes, what are the required conditions to distinguish between electric and magnetic field frequency difference second order coherent processes? The authors should clearly comment on the above questions and on possible alternative mechanisms in a revised manuscript.

Reviewer #3:

Remarks to the Author:

I've worked through the manuscript "Observation of Magneto-Electric Rectification at Non-relativistic Intensities" by M.T. Trinh et al.

In general I find the presentation of the data in general OK. The dynamics itself seems to be in agreement with the scenario the authors describe. If it is a unique effect and not another non-linearity in the system I am not so sure (details see below).

Independent of the source I do not fully see the significance of the observations. I do understand that seeing the ME effect itself is nice but what is missing is showing a functionality. The authors themselves point out some possibilities (optical switching, sensing, energy conversion, THz emission). Some of these effects can be reached using designed (meta-)materials as the authors also explain. These also allow to drive these effects with similar laser powers / field strength as the authors show the present effect. And isn't the granular structure of the pentacene not also simply a "design" and enhancement processes.

The claim that these effects here could be driven as effectively even with incoherent light is only mentioned theoretically in the end.

In summary even with the effect of MER likely to be shown I do not see the big impact without showing a new functionality that was so far not possible without other "designed" systems.

In addition a few remarks to the sections of the manuscript:

Abstract: polycrystalline granular pentacene should be mentioned

Intro on page 2 feels very long and does not conclude what in particular will be shown in the paper. It takes until page 3 to learn what is the actual experimental achievement.

Sample description: What is the influence of the granularity to the data? Is 200nm enough to make the system effectively centrosymmetric? Can the results be shown also for other granularities? Where is the "critical" value?

Experimental setup, Setup characterization: Can you show the SFG scan of the GaAs. How does this scan differ from Fig 2a e.g.

Vice versa how are possible nonlinear effects of SSHG in the measurement of Fig 2a are filtered?

Could there be a SFG contribution from the two incident beams as well (which of course would have a symmetric contribution)?

Page 7: How are the authors sure that no transient changes of the linear optical properties take place that could give rise to the changes in SHG generation? Pumping with 1.55 eV is not so far below gap in particular with these high fluences (what is the actual spot size and resulting fluence in mJ/cm^2). Do 800nm pump/800nm probe or 800nm pump/400nm probe do not show any changes?

Dear Editor,

We wish to thank you for handling our manuscript and thank the reviewers for their comments and suggestions. We have addressed the reviewers' points in detail as outlined below. We have revised the manuscript that incorporated with the reviewers' comments and added three more sections in the supporting information. We believe this manuscript is now ready for publication.

Reviewer 1

This paper reports on the first observation of optical rectification using the magnetic/electric second-order electric-dipole susceptibility. The novelty of the work is that it is the first experimental observation of the effect for non-relativistic intensities. Also, given its large magnitude in centrosymmetric materials, such optical rectification might find many photovoltaic applications. Generally, the paper is well reasoned and I believe that the authors make a strong case that they are indeed observing magnetic/electric optical rectification. The conclusion is based on information gleaned from second harmonic generation as a probe of the static electric field induced by optical rectification. Convincing evidence is gathered from the dependence of the second-harmonic probe intensity on its polarization relative to the pump's propagation direction and its dependence on the pump power.

Based on novelty and the fact that the work is mostly free from errors, I believe that the work should be published provided that the authors respond to the following comments:

Answer:

We thank the reviewer for taking the time to read our paper and for the constructive comments. We are grateful to the reviewer for pointing out the novelty of our work and for suggesting to publish the paper.

I begin with the major questions.

1. While I believe that the authors make a good case for the mixed term $E_i(\omega)H_j^(-\omega)$ as the origin of optical rectification, one could imagine that the electric-only quadrupolar susceptibility could mix two electric fields according to $E_i(\omega)\partial_j E_k^*(-\omega)$ in a centrosymmetric material to also yield a static field in the form of a quadrupolarization. It would give the same results as presented in this paper for a sample that is thinner than a wavelength, as it is in this work. Since the authors are claiming to report on optical rectification that includes the magnetic field, they need to rule out other possibilities such as this one. Please provide a reasoned argument of how you eliminate this contribution.*

Our analysis of the quadrupole interaction shows that it is identically zero in our geometry. This is because the susceptibility vanishes, as we now explain in detail in SI Section 4.

2. Much of the authors' past work relies on a model developed for molecules in a gas or a liquid, where the electrons couple to the nuclei to provide a torque that results in molecular

libration. The authors state that even in a solid where the molecules are rigidly held in place, the same mechanism holds due to librations of the electrons in the fixed molecule over the same time scale as that of a librating molecule. I have scanned through past papers that the authors reference, and I do not see a rigorous calculation of such an effect. Rather, they infer that such an effect takes place based on the fact that they see a comparably large effect in a solid and with the slower time scale, stating that it "... may reflect the timescale of electron librational relaxation taking place in the sample during the torque dynamics initiated by the magneto-electric process." It appears that the mechanisms of the effect in solids is not understood, so the authors should either point to more convincing work that this is indeed the mechanism, or make it more clear that they are speculating on the mechanism.

We agree with the reviewer that a more comprehensive theoretical model for magneto-electric response in solids could be useful. In fact we view the experimental results presented in this paper as potentially providing a guideline for the development of a more quantitative picture of M-E rectification in solids. However, we disagree with the reviewer that there is any speculation on the mechanism responsible for our results.

Our observations and analysis rule out any origin for the pump-induced SHG signal in our experiments other than one driven by EH^* of the incident light. Hence there is no doubt that the mechanism is magneto-electric. On the other hand, the reviewer is correct in the implication that a quantitative theory does not yet exist to explain what factors might be chiefly responsible for the absolute magnitude of ME effects in solids under non-relativistic conditions. The situation is clearly less well-defined than in simple molecular liquids where we have had some success in past work at explaining enhancement of ME phenomena by parametric resonance and magnetic torque. However, the purpose of the present paper is simply to present the first observation of magneto-electric rectification. The experimental work stands on its own. Nevertheless, we have added a statement to clarify that we think ME effects are universally enhanced by parametric resonance and magnetic torque.

3. *Since the large optical rectification response is one of the selling points, it would be useful for the authors to compare their result per molecule with the purely electric one in the best competing system.*

In the present paper we do not address the magnitude of the rectification response, be it "large" or "small". While resonant ME rectification offers some prospect of "large" response according to Fisher, W. M. & Rand, S. C. "Optically-induced charge separation and terahertz emission in unbiased dielectrics" *J. Appl. Phys.* **109**, 1–8 (2011), the current experiments were not performed on resonance. So "large" response is not claimed in the current work which merely reports the first observation of magneto-electric rectification. However, the ME rectification response is significant that we can experimentally observe at a non-relativistic light intensity. In fact we see no reason that ME rectification would exceed all-electric rectification response under comparable conditions. Its novelty lies in extending this type of second-order nonlinearity to centrosymmetric materials. Furthermore, the unique forward direction of the rectification field would provide unforeseen applications in nonlinear optics.

Other issues:

1. The static field induced by optical rectification breaks inversion symmetry and changes the coherence length, so the probe signal should be a convolution of the two and not necessarily a linear function of the pump polarization or the intensity. The authors should mention if this is taken into account and if not, how large of an effect it might have. For example, the data points in Figure 4 seem to be systematically low in the middle of the power range and systematically higher in the high power range. As a result, the power dependence appears greater than quadratic, which could be explained by coherence length changes. With so few points and large error bars, I understand that this may not be statistically significant. Also, I recommend that the authors include a fit to a horizontal line in Figure 4b.

The reviewer is correct that the data in Fig. 4a have large error bars. We have now added a statement to emphasize this in the caption. The low signal-to-noise ratio of the pump-induced SHG signal does not warrant more sophisticated analysis. We also ignored the power dependence of the last term in Eq. (3) of the paper since it was deemed to be negligible as explained in SI. This term and the reviewer's perspective on changes of coherence length could be taken into account in future work at higher powers.

Finally, we have now included a horizontal line fit in Fig. 4b.

2. please state how you determined experimental uncertainty.

The experimental uncertainty was determined by the variance of the data before time zero, as now stated in the caption of Fig.2 and Fig.3

3. Does the fit in Figure 2a use adjustable parameters? If so, how many?

Yes, the fitting procedure used adjustable parameters including the rise and decay times and the amplitude.

4. The polarizations add according to Equation 3 only if the surface contribution is in phase with the volume contribution. Please state the condition under which Eq. 3 holds and then argue why the condition is met in your experiments.

We agree that Eq. (3) can only be applied when the surface and bulk contributions are in phase. In this experiment, both contributions reflect conversion of the same probe wave and therefore have the same phase. Details of the nonlinear intensity analysis are now given in Section 3 of SI.

5. Please state the sample thickness. Though it is given in the supplemental materials, it would be convenient to include in the main paper.

We added the sample thickness to the main text.

6. Why pentacene? It would be useful to know if there is something special about it. Would any organic material with lots of delocalized electrons work?

ME phenomena are universal, meaning that it can be observed in any materials including organics. However we chose pentacene because its first resonance is close to our excitation laser at 800 nm, while it has an absorption tail that does not significantly overlap the excitation wavelength. Pentacene is also a convenient choice for a thin sample that presents little dispersion of the ultrafast pulses. These points are included in the revised manuscript.

I assume that the editors will take care of the grammatical errors and typos, so I will not comments on them. To conclude, this is a nice paper on an interesting topic so should be published. My major questions should be answered since they are at the core of the claims made. The other issues are less critical, but clarify how the analysis is applied and subtleties taken into account that could affect the conclusions.

Again, we are grateful reviewer for the useful comments that definitely improved our manuscript. We read the manuscript carefully for grammatical errors and typos. We incorporated all these comments to improve our manuscript and supporting information.

Reviewer #2 (Remarks to the Author):

The authors propose a nonlinear mechanism arising from both the electric and magnetic field of laser light (EH). This second order nonlinear process is argued to result in magneto-electric rectification even though the magnetic field of laser light is weak for the intensities used. To explain the amplification, the authors make an analogy to molecular systems, where nonlinear magneto-electric interactions have been shown to drive bound electrons on curved trajectories via an optical torque mechanism.

The importance of this paper is that it proposes a possible way to transiently break temporal and spatial inversion symmetry by using an electromagnetic field, rather than breaking inversion symmetry, and thus generate a second order nonlinearity. In particular, a symmetry breaking static field is generated by the second order polarization via a second order frequency difference process that involves both the electric and magnetic fields of light. While such dynamical symmetry breaking process is expected to be weak, since the magnetic field strength is suppressed for the intensities used, the authors make an analogy to previous work in molecular systems to argue that such second-order process is enhanced.

The importance of the proposed effect is that it allows for control of dynamical symmetry breaking by using the laser electromagnetic field. In this way, new physical properties that are directly controlled by light may be expected, for example, in quantum materials, provided that the mechanism of amplification applies to such materials. The possibility for dynamical symmetry breaking via rectification by using the electromagnetic field propagating in thin films has received attention recently in superconductors and topological materials (see, for example,

the works in Nature Photonics 13, 707–713 (2019), Phys. Rev. Lett. 124, 207003 (2020), etc). Another possibility for transient magnetization effects in condensed matter systems, photogenerated through coherent nonlinear processes during the laser pulse despite the laser's weak magnetic field, was suggested in Nature Physics 5, 517 (2009).

We thank the reviewer for taking the time to review our manuscript in detail and for recognizing the importance of our work.

In the present experiment, SHG may be generated at the surfaces of the thin film, but the authors were able to identify two different signals with different behaviors, and thus distinguish between SHG and the proposed EH nonlinear effect that occurs during the pump pulse, as it is pump-induced. In particular, SHG from the probe field was observed to arise due to symmetry breaking by magneto-electric interaction induced by the pump pulse, which gives rise to a rectification field. In my opinion, the authors provide evidence that they observe a pump-induced second-order signal with characteristic polarization and time-delay dependence, which they attribute to the proposed dynamical symmetry breaking mechanism due to the pump electric and magnetic fields that generate a static electric field.

The authors have presented evidence for their proposed mechanism, except for the reason for its enhancement despite the weak laser magnetic field, for which they make the analogy to their previous works in molecular systems. The authors should comment on this, i.e. is this amplification a general effect or does it apply to specific materials. If the latter is the case, which would make the participation of the laser magnetic field plausible, what is the important condition for obtaining the proposed amplification? Does a similar laser-induced dynamical symmetry breaking mechanism apply to quantum materials, which would allow one to coherently control their properties, or is it specific to nonlinear materials like pentacene? If yes, what are the main conditions for amplification and what kind of nonlinearity is required?

Our observations and analysis rule out any origin for the pump-induced SHG signal in our experiments other than one driven by EH* of the incident light. Hence there is no doubt that the mechanism is magneto-electric. The enhancement of ME effect is universal and can be applied to all materials. However, the effect is stronger in some certain conditions such as at or near-resonant excitation, in the materials that support electron delocalization in which the electron movement can be extended. Though no quantitative theory exists as yet to explain the enhancement of ME effects in solids specifically, we believe the enhancement is due to a combination of parametric resonance and torque dynamics that apply to librating electrons in condensed matter. The present paper emphasizes the first observation of magneto-electric rectification and the experimental work stands on its own. Nevertheless, we have added a statement in our conclusions to clarify what we think the origin of ME enhancement is.

We believe that this effect can be observed and enhanced in quantum materials. In some cases, such as metamaterial, the material engineering allows charge to move in a constrained trajectory that could further enhances the nonlinear coupling of electric and magnetic field components of light. (Ref. #11. Klein, *et al.* Second-harmonic generation from magnetic metamaterials. *Science* **313**, 502–504 (2006)).

Could a similar dynamical symmetry breaking effect and light-induced symmetry breaking static field (rectification) occur in a highly nonlinear thin film material with inversion symmetry due to the electric field alone, as proposed e.g. in Nature Photonics 13, 707 (2019), Phys. Rev. Lett. 124, 207003 (2020)? What is the main evidence in the experiment that requires invoking the weak magnetic field of the laser despite the weak Lorentz force?

A centrosymmetry can be broken by a direct current (DC) or by a slowly varying field such as THz (in the above references). However, at the optical frequency, it does not happen. If the rectification field comes from all-electric contribution, and if it is a transverse field (in the XY plane in our geometry, Fig.1), the pump-induced SHG polarization angle dependence is opposite to our observation. E.g. the pump-induced SHG should be strongly depending on the pump polarization angle.

We also have inserted a statement in our paper and detailed analysis in the supplementary information (part 4) to show that an all-electric quadrupolar interaction cannot mediate pump-induced symmetry-breaking in our geometry. This is because the relevant quadrupolar susceptibility elements have paired indices and vanish. Hence our observations are due to the magneto-electric effect. The main evidence is that bulk SHG is forbidden in centrosymmetry and our results are shown to be incompatible with alternative mechanisms such as surface harmonic generation, quadrupolar rectification, electron-hole pair generation and charge separation, etc. The enhancement is universally provided by the means in the answer above.

In conclusion, this paper proposes an all-optical symmetry breaking mechanism that may also open new ways to control quantum materials, if the authors can argue that it is widely applicable, i.e. amplification is possible under general conditions in condensed matter, as well as in molecular systems. Is there an estimate for what kind of nonlinearity of the centrosymmetric material is required for significant symmetry breaking? Is rotational motion as in molecular systems required to amplify the proposed effect and make it observable? What kind of magnetic interaction is required for the proposed effect in different systems? Is this specific to certain thin film systems, and if yes, what are the required conditions to distinguish between electric and magnetic field frequency difference second order coherent processes? The authors should clearly comment on the above questions and on possible alternative mechanisms in a revised manuscript.

These are all good questions, but ones that have been largely answered in our prior publications on related magneto-electric nonlinearities. (1) Laser intensities on the order of 10^8 W/cm² are needed to initiate significant PT-symmetric nonlinearities mediated by the enhanced Lorentz force. (2) Rotational motion of molecules is not needed because parametric resonance can enhance the Lorentz force on its own. But we believe that the librational motion of electrons in solids contributes to further enhancement, just as in simple molecular systems we have studied. (3) Magneto-electric interactions account for this. (4) ME effects are universal since they are not prohibited by spatial inversion symmetry. They are distinguishable from electric effects by polarization angle dependence. See SI sections 3 & 4.

Reviewer #3 (Remarks to the Author):

I've worked through the manuscript "Observation of Magneto-Electric Rectification at Non-relativistic Intensities" by M.T. Trinh et al.

In general I find the presentation of the data in general OK. The dynamics itself seems to be in agreement with the scenario the authors describe. If it is a unique effect and not another non-linearity in the system I am not so sure (details see below).

Independent of the source I do not fully see the significance of the observations. I do understand that seeing the ME effect itself is nice but what is missing is showing a functionality. The authors themselves point out some possibilities (optical switching, sensing, energy conversion, THz emission). Some of these effects can be reached using designed (meta-) materials as the authors also explain. These also allow to drive these effects with similar laser powers / field strength as the authors show the present effect. And isn't the granular structure of the pentacene not also simply a "design" and enhancement processes. The claim that these effects here could be driven as effectively even with incoherent light is only mentioned theoretically in the end. In summary even with the effect of MER likely to be shown I do not see the big impact without showing a new functionality that was so far not possible without other "designed" systems.

We thank the reviewer for taking the time to read and review our manuscript. The magneto-electric rectification reported in this paper is indeed a universal phenomenon that can take place in any dielectric material, in particular lossless, natural optical materials. Thus it avoids the losses and design challenges of metamaterials altogether. There is nothing unique about pentacene other than the proximity of its resonance to the laser frequency and the fact that it can be prepared as a centrosymmetric material which allowed us to clearly distinguish our results from other processes. Thus the applications do not require the design of nano-structured materials and offer the prospect of direct energy conversion of sunlight by ME rectification and switching of light-by-light at ninety degrees which cannot be done by any other means.

In addition a few remarks to the sections of the manuscript:

Abstract: polycrystalline granular pentacene should be mentioned

We now mention the sample in the abstract.

Intro on page 2 feels very long and does not conclude what in particular will be shown in the paper. It takes until page 3 to learn what is the actual experimental achievement.

We modified the abstract and main text accordingly. We added a sentence to the first paragraph of the introduction summarizing the experimental advance.

Sample description: What is the influence of the granularity to the data? Is 200nm enough to

make the system effectively centrosymmetric? Can the results be shown also for other granularities? Where is the “critical” value?

We found no evidence that the granularity of our samples has any effect on the results. Our sample is effectively centrosymmetric as evidenced by no bulk background SHG signal which could not be accounted for as a film surface contribution in the fits of Fig. 3b. Note that when the probe field is parallel to the surface (polarization angle = 0), we do not observe any static SHG signal apart from the noise floor. The average grain size in our sample was corrected to ~100 – 200 nm as we clarify in both the text and section 6 of the SI.

Experimental setup, Setup characterization: Can you show the SFG scan of the GaAs. How does this scan differ from Fig 2a e.g. Vice versa how are possible nonlinear effects of SSHG in the measurement of Fig 2a are filtered? Could there be a SFG contribution from the two incident beams as well (which of course would have a symmetric contribution)?

We include the SFG from GaAs in the supporting information. The SFG from GaAs determines the instrumental response or temporal resolution. It is symmetric with a rise-time of ~0.10-0.15 ps.

SFG does not contribute to the frequency-doubled probe intensity in this work. Our experimental setup utilized a crossed-beam geometry (pump and probe directions at 90°), so that SFG signals, if present, would propagate at 45° with respect to the pump and probe beams. Thus they would not hit the SHG detector located in the probe direction.

To filter out the SSHG, we used a mechanical chopper to modulate the pump light at 1 kHz. The pump-induced signal was measured as signal (ON) – signal (OFF) controlled by a LabVIEW program. To synchronize the laser frequency with the chopper, a fast photodiode was used for triggering. We have added this explanation to the method part.

Page 7: How are the authors sure that no transient changes of the linear optical properties take place that could give rise to the changes in SHG generation? Pumping with 1.55 eV is not so far below gap in particular with these high fluences (what is the actual spot size and resulting fluence in mJ/cm²). Do 800nm pump/800nm probe or 800nm pump/400nm probe do not show any changes?

Transient charges liberated by irradiation by the incident laser could in principle form a space charge field similar to a rectification field if a mechanism existed to enable charge separation. Hence we considered whether 1-photon or 2-photon absorption could account for our results.

The pentacene thin film has a peak absorption at 670 nm, with a tail that extends to 750 nm. The absorption of laser light at 800 nm, the center wavelength of the incident laser, was determined to be negligible. Nevertheless, even a small absorption at 800 nm could generate some electron-hole pairs and conceivably produce a space charge field. However, such a field would have a linear dependence on pump power, whereas we observed a quadratic dependence of the pump-induced harmonic signal on pump power.

One could also argue that the material undergoes 2-photon absorption. In this case, the dependence of the pump-induced field would be quadratic with respect to the pump power, similar to that of an MER field. However, the signal rise time would be as fast as the pulse duration and its decay time would have to be either that of the singlet exciton lifetime of 24 ns or the singlet fission time scale of 80 fs (Refs. 29-30, main text). However, our rise time was ~ 0.5 ps which is five times the pulse width and the signal decay time was ~ 0.6 ps, which is very different from exciton timescales. Hence we discarded this possibility, but included a discussion of this issue in section 5 of the SI.

The pump diameter at the sample position was 1.5 mm, however, the laser spot on the sample was an elliptical shape with axes of 1.5 and 2.1 mm because the incident angle was 45° . For 100 fs pulses, our maximum pump pulse intensity on the sample was therefore $\sim 7 \times 10^8$ W/cm². Our experimental setup is unique that uses pulse-front tilt for the pump and probe pulses in a cross-beam geometry. This design was used to overcome the poor temporal resolution by the crossbeam pump-probe geometry. Both pump and probe wavelengths were set at 800 nm. The tunable wavelength has not been implemented in our setup, and it is not trivial for the pulse front tilt system.

The experimental beam size and intensity values were also added to Methods.

Reviewers' Comments:

Reviewer #1:

Remarks to the Author:

See attached pdf file

Reviewer #2:

Remarks to the Author:

The authors have answered the main questions posed by the three referees. The remaining question is how the proposed mechanism works in a solid, but without proper theory, the authors are using the analogy to molecular systems. I recommend publication as a first observation paper.

Reviewer #3:

Remarks to the Author:

Dear Editor,

I have worked through the revised manuscript. They have answered all questions and comments and I agree with their statements. I still miss seeing some of the new functionalities in this paper that would show the big impact. But here they show that the MER effect indeed exists and given their new and extended explanations I agree that the observations indeed are due to this. If that effect really will become important and allow for new devices etc. future studies have to show (as also reviewer 2 concludes).

The challenges from concept to new functionality/device are still not fully discussed.

Nevertheless the observation of the new MER effect seems sound and therefore publication is my opinion is fine.

We would like to thank all reviewers for carefully working on our manuscript and for recommendation. In the following please find our response to the reviewers' comments (in blue).

Reviewer # 1

I am satisfied with the Authors' response to the minor issues. However, I am still concerned that the purely electric term $E\nabla E$ contributes and that the argument in Section 4 of the Supplemental Information is not correct. There are two issues, as follows. First, the correct polarization for the quadrupolar contribution is of the form $P_i = \chi_{ijkl} E_j \partial_k E_l$. Then for an electric field along x, a polarization would result along z from the term $P_z = \chi_{zxzx} E_x \partial_z E_x$. Secondly, it is not true that for an isotropic material, a particular tensor component must be invariant upon rotation. If this argument were true, the linear susceptibility χ_{ii} would vanish because a rotation about z of χ_{xx} by angle θ would give $\chi_{xx} \cos^2 \theta$, where θ is the angle between x and i. The correct expression for rotation is given by $\chi_{ii} = \chi_{xx} \cos^2 \theta + \chi_{yy} \sin^2 \theta$. If the material is isotropic, then $\chi_{xx} = \chi_{yy}$ making χ_{ii} invariant upon rotation. In analogy to the authors' simple model of a free electron in an electric and magnetic field, I could formulate a simple model of confined charges in a molecule that also leads to optical rectification. Since the electron does not move freely in a molecule, the purely electric model might be more plausible. If the authors agree that the quadrupole electric contribution is possible, it would be useful to see a hand-waiving argument of why it would be small compared with the electric/magnetic term for their system. If they find that it is not necessarily small, then they need to state that quadrupole electric polarization is a possibility. If my argument is in error, then of course this paper should be published as is.

Answer: We thank the reviewer for pointing out the errors. We have incorporated the reviewer's suggestions and modified the supporting information section 4. We pointed out that the quadrupolar contribution to rectification in z-direction is negligible and therefore cannot be accounted for our observation. The following has been added in the supporting information.

"In addition to magneto-electric interactions, quadrupolar electric interactions can theoretically give rise to second order nonlinear response. Consequently, it might be thought that such an interaction could explain rectification in the experiments reported in this work. However, this is

not the case. Quadrupole interactions can lead to frequency-doubling but not to rectification, as we show below. Consider an x-polarized pump field $E_x = E_{x0}e^{i(\omega t - kz)} + c. c.$. A second order polarization that points along the pump propagation axis (z) and results from a quadrupole interaction [3] then has the form

$$P_z^{(q)} = \epsilon_0 \chi_{zzzx}^{(q)} E_x \frac{\partial}{\partial z} E_x$$

In an isotropic material the tensor susceptibility element $\chi_{zzzx}^{(q)}$ does not vanish. However, upon substitution of the pump field, the nonlinear polarization is found to be

$$\begin{aligned} P_z^{(q)} &= \epsilon_0 \chi_{zzzx}^{(q)} \{ (E_{x0} e^{i(\omega t - kz)} + c. c.) (-ik E_{x0} e^{i(\omega t - kz)} + c. c.) \} \\ &= \epsilon_0 \chi_{zzzx}^{(q)} \{ -ik E_{x0}^2 e^{2i(\omega t - kz)} + c. c. \} \end{aligned}$$

This nonlinear polarization consists exclusively of second harmonic terms. The static field terms vanish. Hence the quadrupole interaction does not support rectification. Moreover, harmonic radiation from this interaction yields no SHG signal from the probe alone because the polarization cannot radiate along the propagation axis of the probe, the direction in which the detector is located. While quadrupolar SHG from the pump alone could reach the detector directly, it would not give rise to a pump-induced change in SHG from the probe rather to the total background. Therefore quadrupole interactions cannot account for the pump-induced harmonic probe signal (Δ SHG) in our experiments.”

Reviewer #2

The authors have answered the main questions posed by the three referees. The remaining question is how the proposed mechanism works in a solid, but without proper theory, the authors are using the analogy to molecular systems. I recommend publication as a first observation paper.

We thank the reviewer for the recommendation.

Our experimental observation would be a great guideline for the development of a comprehensive theoretical model for magneto-electric interaction in solid state.

Reviewer #3

I have worked through the revised manuscript. They have answered all questions and comments and I agree with their statements. I still miss seeing some of the new functionalities in this paper that would show the big impact. But here they show that the MER effect indeed exists and given their new and extended explanations I agree that the observations indeed are due to this. If that effect really will become important and allow for new devices etc. future studies have to show (as also reviewer 2 concludes).

The challenges from concept to new functionality/device are still not fully discussed.

Nevertheless the observation of the new MER effect seems sound and therefore publication in my opinion is fine.

We thank the reviewer again for carefully working on our manuscript and for the recommendation.

Magneto-electric interaction is a universal phenomenon and it is a part of fundamental light-matter interaction. The observation of the ME nonlinearities is itself important as scientists may need to take into account this phenomenon in studying/analyzing nonlinear light-matter interaction. We have foreseen some applications of the magneto-electric interactions as presented in the main manuscripts. For example: THz emission in a centrosymmetric system without an applied bias has been thought not to be possible but now with the ME interaction it can be realized.

Reviewers' Comments:

Reviewer #1:

Remarks to the Author:

With the errors corrected and the electric quadrupole interactions taken into account, I believe that the manuscript is interesting and correct so ready for publication.

Mark G. Kuzyk